# Structural Relationship between Psychological Needs and Sport Adherence for Students Participating in Physical Education Class

**Juan M. García-Ceberino** [1,2] , **Sebastián Feu** [1,3] , **María G. Gamero** [3] **and Sergio J. Ibáñez** [3,4,*]

1   Universidad de Extremadura, Facultad de Educación y Psicología, Avenida de Elvas s/n,
    06006 Badajoz, Spain; jmanuel.jmgc@gmail.com (J.M.G.-C.); sfeu@unex.es (S.F.)
2   Faculty of Education, Psychology and Sports Science, University of Huelva, 21007 Huelva, Spain
3   Optimization of Training and Sports Performance Research Group (GOERD), Faculty of Sports Science,
    University of Extremadura, 10003 Cáceres, Spain; mgamerob@alumnos.unex.es
4   Faculty of Sports Science, University of Extremadura, 10003 Cáceres, Spain
*   Correspondence: sibanez@unex.es

**Abstract:** The satisfaction of basic psychological needs leads students to engage in a sport modality on their own initiative. In the Spanish public educational system, mixed and heterogeneous, the gender and sport experience of students influence the teaching and motivation of invasion sports. This study investigated whether students' gender and sport experience, and model influence the psychological variables (basic psychological needs and sport adherence) when teaching school soccer and basketball. Furthermore, correlations were calculated between these psychological variables. The study involved 165 fifth and sixth grade students (Mage, $11.27 \pm 0.68$ years old) from several Spanish state schools in the same autonomous community. A non-random convenience sample was used. The needs for autonomy, perceived competence and social relationships were measured using the Basic Psychological Needs in Physical Exercise Scale. Sport adherence was measured using the Measure of Intentionality to be Physically Active. A Confirmatory Factor Analysis, Cronbach's Alpha, Average Variance Extracted and Composite Reliability were used to analyze the psychometric properties of the scales. Descriptive and inferential analyses were obtained after performing a Mixed Linear Model and a Bonferroni Post Hoc. There were significant differences in the autonomy need according to the students' gender (boys > girls). In addition, boys who learned with the TGA model scored significantly higher than girls on this need. Attending to sport adherence, there were significant differences in favor of experienced students. Likewise, a regression analysis (structural equation model) revealed that the autonomy need showed low association with sport adherence. Higher perceived competence ($\beta = 0.52$) and social relationships ($\beta = 0.36$) were associated with greater adherence to sport ($R^2 = 0.65$). Teachers' Knowledge and proper managing basic psychological needs will have educational, social and health benefits, as it will increase the likelihood of physical activity in and out of school.

**Keywords:** invasion sport; gender; primary school; sport experience; structural equation

## 1. Introduction

School physical education (PE) for the initiation of invasion sports is a privileged scenario for the promotion of physical activity (PA) [1]. The Self-Determination Theory (SDT) [2] is a theory of human motivation and personality that emphasizes the importance of the individual's ability to self-regulate his or her behavior, that is, it understands that individuals are responsible for their actions and relates self-determined behaviors to the propensity to engage in a wide range of human activities [2]. It comprises four mini-theories, including the Basic Psychological Needs (BPNs) Theory [3]. The BPNs theory involves three basic needs for the individual to develop and thrive: (1) autonomy (i.e., the possibility of voluntary choice in one's own activities and goals through participation in

the decision-making process); (2) perceived competence (i.e., the ability to participate and engage in activities, and the execution (effectiveness) of the activities to achieve goals and success); and (3) social relationships (i.e., the need to reinforce the feeling of acceptance and belonging to the community in which one lives) [3].

Interventions based on the SDT could be useful in the process of promotion and adherence to PA [4]. Thus, the satisfaction of three BPNs by students or players leads to an increase in intrinsic motivation, the most self-determined regulation, which causes students or players to engage in a sport modality on their own initiative (e.g., involvement in a sport for pleasure, fun, etc.) [5]. However, the frustration of these BPNs leads to an increase in extrinsic motivation (e.g., involvement in a sport to obtain a reward or avoid punishment) or demotivation (which can lead to the abandonment of sport practice) [5]. Therefore, the motivation generated by the sport practice is one of the reasons that lead students or players to get involved in, or on the contrary, to abandon the sport practice [6]. In PE, García-Ceberino, et al. [1] report that the more competent the students feel in school soccer, the greater the degree of adherence shown by them. In turn, Lamoneda and Huertas-Delgado [7] report that students perceive a lower satisfaction of the BPNs as they advance in level and educational stage, producing a decrease in sport practice. In this regard, PE teachers have a decisive role in the promotion of sport and adherence in and out-of-schools [8]. In has regard, it has been shown that the teacher's adoption of a controlling interpersonal style is closely related to less self-determined regulations of motivation, thus generating a negative environment conditioned by the students' lack of autonomy [9].

In the field of Sports Pedagogy, there are different models to teach invasion sports during PE classes [10]. Given its proven benefits, teachers should implement a game-centered approach that facilitates the enhancement of motor learning, technical skills and the development of the tactical cognition process of the sport in question, moving away from direct instruction excessively focused on teaching technical skills [11]. Therefore, this approach focuses attention on the student through tasks that stimulate the intertwined development of technical skills and tactical awareness [11]. Within the game-centered approaches, the Tactical Games Approach (TGA) [12] is suitable for the teaching of invasion sports. The TGA is an adaptation of the Teaching Games for Understanding (precursor of the game-centered approach) [13], and is characterized by the use of modified games and/or small-sided games to facilitate the understanding of the game [12]. They allow students or players to experiment and solve tactical problems by modifying or adapting the length of the field, the number of players, the rules of each selected game, etc. [14]. Specific formative programs have been intentionally designed and validated for the teaching of invasion sports, such as soccer [15,16] and basketball [17–19]. These programs are oriented to primary PE and based on different models: TGA and traditional direct instruction.

From a pedagogical perspective, in addition to the model applied, it is relevant to take into account the gender and initial experience of the students when planning sport teaching in PE, since both factors influence the teaching-learning process when working with mixed and heterogeneous school groups. This fact has been demonstrated in previous research on soccer [20–22] and basketball [23,24] in primary schools. However, most of the research related to the methodological aspect does not usually consider the effect of both factors. In these aforementioned studies, the individual response of the class-groups (random factor) was not controlled.

The lack of interest of students in PE classes during primary school is increasing [25]. One of the reasons for this lack of interest/motivation could be linked to the model applied by the teacher, who in turn should consider the heterogeneity of the group-classes. This type of studies contribute of students contribute to the sustainability of the educational, social and health systems, since they provide knowledge of the influence of psychological variables for the promotion of PA in and out of school [4], with health benefits by avoiding sedentary lifestyles and their consequences. Thus, this study aimed to analyze whether students' gender and sport experience, and teaching model influence the psychological variables (BPNs and adherence to sport) when teaching school soccer and basketball. More-

over, correlations between these psychological variables were calculated. We hypothesized that: (1) Boys would report higher BPNs satisfaction and adherence to sport than girls; (2) Experienced students would report higher BPNs satisfaction and adherence to sport than non-experienced students; (3) The BPNs would be correlated with the intention to practice PA (adherence to sport); and (4) The perceived competence is the need that would report the highest correlation with the degree of adherence to sport.

## 2. Materials and Methods

### 2.1. Study Design

A quasi-experimental study was carried out [26]. Different teaching programs on soccer [15,16] and basketball [17–19] were applied in the primary schools. These programs lasted three months (April to June), with one or two sessions per week (11 practical sessions in total). The school authorities indicated the days when researchers could go to the schools. It was also a cross-sectional study [26], as data collection took place on a specific day at the end of the teaching programs. The class-groups of the participating primary schools were not modified, thus maintaining the ecological validity of the study.

Despite the fact that the study did not require invasive measures to obtain the data, the University Ethics Committee approved its protocol [protocol code: 105/2022]. Authorization was also requested from the primary schools and the PE teachers at these schools. Moreover, the school board within the school curriculum approved the implementation of the study.

### 2.2. Participants and Procedure

The study involved 165 fifth and sixth grade students (Mage, 11.27 ± 0.68 years old), including 78 boys (47.30%) and 87 girls (52.70%), from five Spanish public primary schools in the autonomous community of Extremadura. In this regard, a non-random convenience sample was used, since the samples were selected according to two aspects: attendance at the schools where the academic authorities authorized their participation, and the proximity of the researchers' residence to these schools (about 30 km). The class-groups in the Spanish educational system are mixed and heterogeneous.

The inclusion criteria for participation in the study were: (1) the parents/legal guardians had to have signed an informed consent; (2) participants had to have attended at least 80% of the sessions; and (3) they had to have responded adequately (i.e., all items) to the instruments.

Information on the schools and class-groups participating in this study are shown in Table 1, clarifying the professional and the teaching model used with each school and class-group.

**Table 1.** Information on the participating schools and class-groups.

| School | Class-Group | Students | Sport | Professional | Model Used |
|---|---|---|---|---|---|
| School 1 | [1] 5th grade A | 20, 8 boys | Soccer | PhD. and soccer license | Tactical Games |
| | [1] 5th grade B | 21, 15 boys | Soccer | PhD. and soccer license | Direct Instruction |
| School 2 | [2] 6th grade A | 17, 6 boys | Basketball | PhD. and basketball license | Direct Instruction |
| | [2] 6th grade B | 19, 9 boys | Basketball | PhD. and basketball license | Tactical Games |
| | [3] 6th grade C | 18, 8 boys | Basketball | In-service teacher | Mixed, closer to DI |
| | [3] 6th grade D | 19, 8 boys | Basketball | In-service teacher | Mixed, closer to DI |
| School 3 | [2] 6th grade A | 13, 8 boys | Basketball | PhD. and basketball license | Tactical Games |
| School 4 | [3] 5th grade A | 16, 8 boys | Basketball | In-service teacher | Mixed, closer to DI |
| School 5 | [3] 6th grade A | 22, 7 boys | Basketball | In-service teacher | Mixed, closer to DI |

Note: PhD. = Doctor in PA and Sport; DI = Direct Instruction. [1] A professional from outside the school taught the specific soccer programs. [2] A professional outside the school taught the specific basketball programs. [3] The school's own PE teacher designed and taught a didactic unit (learning situation) based on his or her experience and pedagogical knowledge.

The soccer [15,16] and basketball [17–19] programs taught by researchers from outside the school, based on the TGA and DI models (Table 1), were specifically designed for teaching these invasion sports in PE. In addition, they were validated by an expert panel with excellent content validity and internal consistency values. For each invasion sport, the programs differed in model (TGA/DI), but they were similar in specific content and game phases. In turn, the learning situations designed by the in-service teachers were based on a mixed model, closer to the traditional DI model. The choice of the class-groups to participate in each formative program/learning situation was randomly selected. Both invasion sports contain common tactical principles, among others: maintaining possession of the ball or the advance towards the opponent's goal [27].

At the end of the application of the specific teaching programs and learning situations in the different schools, the students answered two instruments, in the Spanish version adapted to PE: (1) the scale for the Measurement of Basic Psychological Needs in Physical Exercise-BPNS-PE (four items for each need) [28]; and (2) the Measure of Intention to be Physically Active-MIPA (five items) [29]. Both instruments were answered using a Likert-type scale from 1 (strongly disagree) to 5 (strongly agree). The approximate response time was 25–30 min. The main researchers were present to explain the instruments and answer any questions.

Finally, once all the data were collected, they were exported to the statistical software for descriptive and inferential analysis.

### 2.3. Statistical Analysis

Descriptive data were calculated, using the mean and standard deviation, to characterize students' scores on the BPNs and sport adherence. The data presented a non-normal distribution after calculating the Kolmogorov-Smirnov test, indicating the use of non-parametric tests. A transformation from non-parametric to parametric data was then performed [30]. A Mixed Linear Model and Bonferroni Post Hoc [31] were performed to identify statistical differences between the variables studied. In this model, the individual response of each class-group (random factor) was controlled to analyze whether the continued response of the same group-class affected the results. A regression analysis (structural equation model) was also performed to verify the predictive capacity of the BPNs for the intention to practice PA (adherence to sport). Statistical analysis was performed with SPSS, version 27 (IBM Corp. Released 2020. IBM SPSS Statistics for Windows, Version 27, IBM Corp, Armonk, NY, USA) and Jamovi software, version 2.3.24 [32]. Differences were considered significant when $p \leq 0.05$.

### 2.4. Endpoints

An acceptable model fit was obtained with the instruments grouped together (Table 2), according to the values proposed by Hu and Bentler [33]. The AMOS plugin (for SPSS 27.0 statistical software) was used for the analysis [34]. It was not necessary to remove any item from the instruments.

Reliability was calculated using the Cronbach' Alpha ($\alpha$), Average Variance Extracted (*AVE*) and Composite Reliability (*CR*). In this regard, Cronbach's alpha was good for both instruments: BPNS-PE $\alpha = 0.81$; MIPA $\alpha = 0.81$ [35,36], exceeding the minimum acceptable value of 0.60 required by the literature [37]. *AVE* and *CR* were also appropriate for both instruments: BPNS-PE (*AVE* = 0.52; *CR* = 0.69) and MIPA (*AVE* = 0.58; *CR* = 0.80), exceeding the minimum acceptable values of 0.50 for the *AVE* and 0.60 for the *CR* [37].

**Table 2.** Confirmatory Factor Analysis grouping both instruments and independently.

| Measure | BPNS-PE | MIPA | Grouped Instruments | Threshold (Grouped) | Interpretation (Grouped) |
|---|---|---|---|---|---|
| *CMIN* | 68.988 | 5.172 | 123.698 | - | - |
| *DF* | 51 | 5 | 62 | - | - |
| *CMIN/DF* | 1.353 | 1.034 | 1.995 | 1 and 3 | Excellent |
| *CFI* | 0.960 | 0.999 | 0.910 | 0.90 and 0.95 | Acceptable |
| *TLI* | 0.948 | 0.999 | 0.887 | <0.90 | Improvable |
| *SRMR* | 0.068 | - | 0.093 | 0.10 and 0.08 | Acceptable |
| *RMSEA* | 0.046 | 0.014 | 0.078 | 0.08 and 0.06 | Acceptable |
| *PClose* | 0.56 | 0.611 | 0.014 | 0.01 and 0.05 | Acceptable |

Note: BPNS-PE = Measurement of Basic Psychological Needs in Physical Exercise; MIPA = Measure of Intention to be Physically Active (sport adherence); *CMIN/DF* = Chi-Square ratio over Degrees of Freedom; *CFI* = Comparative Fit Index; *TLI* = Tucker Lewis Index; *SRMR* = Standardized Root Mean Squared Residual; *RMSEA* = Root Mean Square Error of Approximation.

## 3. Results

Table 3 shows the inferential results of psychological variables analyzed. There were statistical significant differences in the autonomy according to the gender of the students (boys' > girls' scores; $t = 1.930$; $SE = 0.947$; $p$ bonferroni = 0.05). Furthermore, there were significant differences in this need by gender of the students (boys' > girls' scores; $t = 3.277$; $SE = 1.580$; $p$ bonferroni = 0.02) who learned with the TGA model. On the other hand, there were statistical significant differences in adherence to sport according to the sport experience (experienced students > inexperienced students; $t = 3.23$; $SE = 1.19$; $p$ bonferroni = 0.00).

**Table 3.** Inferential results of psychological variables analyzed.

| Study Variables | BPNs | | | | | | Sport Adherence | |
|---|---|---|---|---|---|---|---|---|
| | Autonomy | | Perceived Competence | | Social Relationships | | | |
| | *F* | *p* | *F* | *p* | *F* | *p* | *F* | *p* |
| Gender | 3.815 | 0.05 * | 3.288 | 0.07 | 1.105 | 0.30 | 2.185 | 0.14 |
| [1] Sport experience | 0.981 | 0.32 | 1.161 | 0.28 | 3.225 | 0.08 | 10.806 | 0.00 * |
| Teaching model | 0.101 | 0.90 | 0.370 | 0.69 | 0.038 | 0.96 | 0.545 | 0.58 |
| Gender * Sport experience | 0.022 | 0.88 | 1.083 | 0.30 | 1.204 | 0.27 | 1.797 | 0.18 |
| Gender * Teaching model | 3.3115 | 0.04 * | 0.146 | 0.86 | 0.575 | 0.56 | 0.046 | 0.96 |
| Sport experience * Teaching model | 2.377 | 0.10 | 0.894 | 0.41 | 0.679 | 0.51 | 1.671 | 0.19 |
| Gender * Sport experience * Model | 0.022 | 0.98 | 0.482 | 0.62 | 0.245 | 0.78 | 0.245 | 0.78 |

Note: *F* = Mixed Linear Model; BPNs = Basic Psychological Needs. [1] It refers to the sport practice in and out-of-school context. * $p \leq 0.05$.

In the BPNs, the Intraclass Correlation Coefficient was close to 0; thus, the responses of each group-class was stable and had no random effect. In contrast, in sport adherence, the Intraclass Correlation Coefficient was not close to 0; therefore, the responses of each group-class was not stable and had random effect (Table 4).

The proposed model (Figure 1) showed that the perceived competence ($\beta = 0.52$) and social relationships ($\beta = 0.36$) correlated significantly with the adherence to sport, obtaining a high coefficient of determination ($R^2 = 0.65$). Therefore, both needs predicted 65% commitment to sport adherence. The perceived competence was the main predictor factor. This percentage of commitment was reported by eliminating the autonomy need from the model because it showed a low correlation with sport adherence.

**Table 4.** Control of the random effect of each group-class.

| | Phychological Variable | $R^2$ Conditional | BIC | ICC | LRT | p |
|---|---|---|---|---|---|---|
| BPNs | Autonomy | 0.233 | 1035.034 | 0.157 | 14.5 | <0.001 * |
| | Perceived competence | 0.155 | 1068.085 | 0.089 | 6.60 | 0.01 * |
| | Social relationships | 0.110 | 1074.732 | 0.069 | 4.35 | 0.04 * |
| Sport adherence | | 0.122 | 1109.311 | $7.55 \times 10^{-16}$ | 0.00 | 1.00 |

Note: BPNs = Basic Psychological Needs; *BIC* = Bayesian Information Criterion; *ICC* = Intraclass Correlation Coefficient; *LRT* = LRT test for Random Effect. Intercept | Class-group. * $p \leq 0.05$.

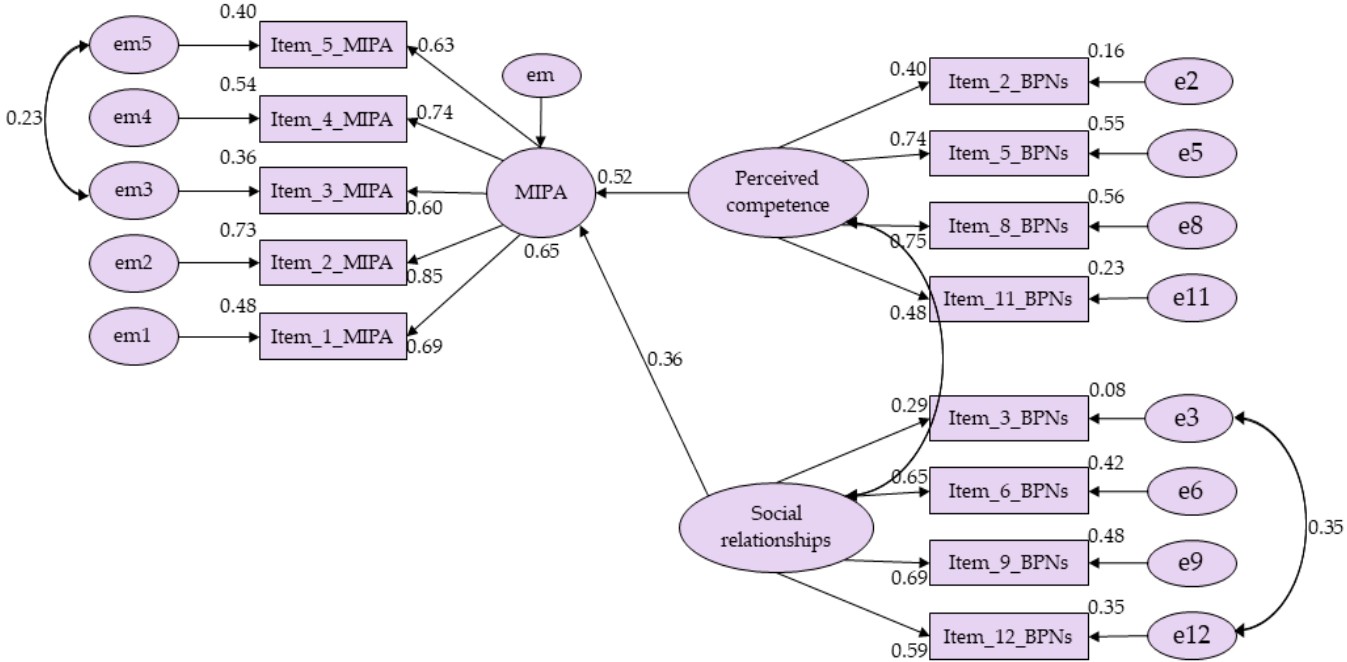

**Figure 1.** Structural equation model in primary students. Note: BPNs = Basic Psychological Needs; MIPA = Sport adherence.

## 4. Discussion

The satisfaction of the BPNs increases students' intention to be physically active [38]. The heterogeneity of PE classes in the Spanish educational system could mean that BPNs satisfaction is not perceived equally within the group of students. The purpose of this study was to analyze whether students' gender and experience in sport, and teaching model influence the various psychological variables (BPNs and sport adherence) when teaching school soccer and basketball. In addition, correlations between these psychological variables were calculated. The main results showed that there were significant differences in the autonomy need according to the students' gender (boys > girls). In addition, boys who learned with the TGA model scored significantly higher than girls in this need. Attending to sport adherence, there were significant differences in favor of experienced students. Therefore, hypotheses 1 and 2 were partially accepted because there were no significant differences in all needs and comparison variables. On the other hand, the perceived competence and social relationships predicted 65% of adherence to sport, with the perceived competence as the main predictor. For these reasons, hypothesis 3 was partially accepted since autonomy showed a low correlation, and hypothesis 4 was fully accepted.

The participating boys reported greater autonomy in the invasion sports of soccer and basketball. In contrast to this study, García-Ceberino, et al. [1] reported that gender influenced perceived competence and students' intentionality to continue playing soccer in and out-of-school, in favor of boys. Gender-based analysis in another study [39] reported

higher BPNs satisfaction and higher intention to be physically active associated with boys (PE students). Likewise, Gutierrez and García-López [40] reported stereotypical participation, mainly in invasion sports, with girls being relegated to a spectator role. In contrast, Fernández-Hernández, et al. [41] reported no significant differences in satisfaction with the three BPNs or in the intention to be physically active according to the gender of the primary students.

Experienced students reported greater sport adherence in the invasion sports of soccer and basketball; however, there were no significant differences in three BPNs. A previous study [1] reported that the initial experience influenced the students' interest in continuing to play soccer in and out-of-school, but it did not influence their perceived competence. On the contrary, Heredia-León, et al. [42] reported that students who participated in extracurricular activities (i.e., out-of-school) were associated with self-determined profiles. The continuous response of each group-class was not stable and had a random effect. It could be said that students' sport adherence varies according to the educational level in which they are enrolled [7], or the most popular/practiced sport in a particular locality. Rodríguez, et al. [43] noted the importance of taking into account the technical differences and motor condition of each student in PE classes. In addition, adolescents girls reported greater barriers to PA practice compared to their boy classmates [44]. Based on our results, PE teachers should promote more self-determined regulations during the teaching-learning of school soccer and basketball, with especial attention on girls and inexperienced students.

In contrast to the study developed by García-Ceberino, et al. [1], on school soccer, there were significant differences in the autonomy need in favor of boys who learned with the TGA model. The characteristics of TGA [45], as opposed to traditional models of DI, favor the autonomy of the students because their efforts are oriented towards success and positive possibilities in the resolution of the tactical problems posed and decision making [46]. In PE, the role of the teacher has been shown to be determinant in students' satisfaction of BPNs [47]. He/she can condition students' behavior through the model employed; creating an environment that generates greater motivation and self-determined regulation. In this regard, Contreras, et al. [48] reported that controlling models focused on technical skills, as opposed to models focused on understanding the game (e.g., the TGA), lead to lower motivation levels in students. Therefore, it is necessary to encourage students' motivation through attractive games (tactical problems) that generate active participation, fun and decision making, in order to obtain significant learning.

The model proposed in our study showed that perceived competence and social relationships were significantly and highly correlated with adherence to the soccer and basketball sports. The autonomy need had a low correlation possibly because students who learned with models focused on technical skills, due to their characteristics (e.g., teacher's controlling style) [9,49], had less autonomy when executing the learning tasks. It is suggested to add strategies based on autonomy support, as it can ensure adherence to sport practice [50]. In this regard, TGA supports autonomy. A previous study [39] reported correlations between the BPNs satisfaction and the intention to be physically active in line with SDT. In addition, perceived competence was the strongest predictor of adherence to sport in this study. In this regard, Lamoneda and Huertas-Delgado [7] indicated that, of the BPNs, perceived competence was also the most positively predictive of PA. A similar study [1], on school soccer, reported that the more competent students feel, the greater their adherence to this sport. In a study with adolescents [51], perceived competence was also significantly associated with their participation in organized sports. Therefore, interventions based on SDT [2], supported by BPNs, could be a useful didactic strategy to motivate primary students to practice PA in other contexts.

Among the limitations of this study, it should be noted that the sport type and the educational stage of the students could cause the results to vary; in fact, the response of each group-class was not stable and had a random effect. Therefore, these results cannot be generalized to sports that do not have the same internal logic as soccer and basketball (e.g., individual sports) nor to middle school or high school. The findings reported the

importance of satisfying the BPNs, especially perceived competence, to improve participation rates in invasion sports with educational (e.g., learning technical skills and tactical awareness), social (e.g., participation in out-of-school sports-recreational activities), and health (e.g., active lifestyle) benefits. In turn, Fernández-Espinosa, et al. [4] indicated that the involvement of families could also influence the promotion of PA in children and adolescents, therefore, this should be taken into account in future research.

## 5. Conclusions

The results of the study indicate that interventions based on SDT, supported by the BPNs, are necessary to motivate primary school students to practice PA in different contexts. In this regard, it is important that students feel competent in the sport in question, in order to increase their intention to be physically active. On the other hand, the heterogeneity of PE classes is a factor to take into account when planning the teaching of soccer and basketball sports, paying special attention to girls and inexperienced students. Knowing and managing BPNs correctly will have educational, social and health benefits by increasing the likelihood of PA in and out of school.

**Author Contributions:** Conceptualization, J.M.G.-C. and S.F.; methodology, J.M.G.-C. and S.F.; formal analysis, J.M.G.-C. and S.F.; investigation, J.M.G.-C., S.F., M.G.G. and S.J.I.; data curation, S.F.; writing—original draft preparation, J.M.G.-C.; writing—review and editing, S.F., M.G.G. and S.J.I.; visualization, S.F., M.G.G. and S.J.I.; supervision, S.F. and S.J.I.; funding acquisition, S.J.I. All authors have read and agreed to the published version of the manuscript.

**Funding:** This study has been partially subsidized by the Aid for Research Groups (GR21149) from the Regional Government of Extremadura (Department of Economy, Science and Digital Agenda), with a contribution from the European Union from the European Funds for Regional Development. The author J.M.G.-C. was supported by a grant from the Universities Ministry of Spain and the European Union (NextGenerationUE) "Ayuda del Programa de Recualificación del Sistema Universitario Español, Modalidad de ayudas Margarita Salas para la formación de jóvenes doctores" (MS-01).

**Institutional Review Board Statement:** The study was conducted in accordance with the Declaration of Helsinki and approved by the ethics committee of the University of Extremadura (approval number: 105/2022; 29 June 2022).

**Informed Consent Statement:** Written informed consent was obtained from parents or legal guardians of the students who were involved in the study.

**Data Availability Statement:** Data will be available upon reasonable request from the corresponding author.

**Acknowledgments:** The authors thank the academic authorities of the schools, the physical education teachers and the students for their participation in the study.

**Conflicts of Interest:** The authors declare no conflict of interest.

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
