# Peer review of "Structural Relationship between Psychological Needs and Sport Adherence for Students Participating in Physical Education Class"

_sustainability, doi:10.3390/su15129686_

Round 1

Reviewer 1 Report

Thank You for giving me the chance to read this interesting paper. I have some (hope - helpful) suggestions about design/analyses and report.

1. I suggest series of mixed (multilevel) linear models instead of Mann-Whitney and Kruskall-Wallis tests. Your data is clustered at school/class, and both (MW and KW) tests used in Your analyses assume independence of observations. 

2. Analogous strategy (in my opinion) should be applied to answer the second question about correlations between psychological issues. The SEM model probably is not accurate due to violation one of the most fundamental assumption of indepndence of observations.

3. Results of regression analysis are not clearly presented  - I don't undestand presented numbers. Are they unstandardised or standardised regression weights or (possibly) partial correlations? What's the standard error (or confidence) of this estimates? Why authors present some colorfull and (meritoriclly) not well design diagram instead of diagram (as I understand) drown in AMOS? I believe that even table of coefficients would be better and more descriptive for any reader.

4. In introduction and discussion authors say a lot about teaching methods but, as I see - they do not explain differences between students in BPNs... I don't see exactly why teaching methods take so much place in the paper if results do not support their importance?

Generally - I suggest to reanalyse Your results, possibly diving into the MLM models (where e.g. independence of observations is not expected) could a) give You one methodology to answer all Your questions and b) give You some better understanding of "weights" of Your demographic and psychological variables in sport adhendance.

Author Response

We would like to express our gratitude to reviewer 1 for the time in reviewing our manuscript and for providing us comments helpful to improve this manuscript quality. We have answered their concerns (all corrections were marked in red and/or change control).

--------------------

Reviewer’ note: I suggest series of mixed (multilevel) linear models instead of Mann-Whitney and Kruskall-Wallis tests. Your data is clustered at school/class, and both (MW and KW) tests used in your analyses assume independence of observations.

Authors’ response: We agree with you. A Linear Mixed Model (random factor = group-class) and Bonferroni Post Hoc were performed (see Statistical Analysis section).

--------------------

Reviewer’ note: Results of regression analysis are not clearly presented - I do not understand presented numbers. Are they unstandardized or standardized regression weights or (possibly) partial correlations? What is the standard error (or confidence) of this estimates? Why authors present some colorful and (meritoriclly) not well design diagram instead of diagram (as I understand) drown in AMOS? I believe that even table of coefficients would be better and more descriptive for any reader.

Authors’ response: Based on your questions, the diagram provided by AMOS software was inserted (Figure 1), giving more clarity to the results.

--------------------

Reviewer’ note: In introduction and discussion, authors say a lot about teaching methods but as I see - they do not explain differences between students in BPNs... I don't see exactly why teaching methods take so much place in the paper if results do not support their importance?

Authors’ response: Now, the new results obtained from the Linear Mixed Model do support the importance of teaching models. This importance has been discussed (Lines 258 to 270).

Reviewer 2 Report

Thank you for the opportunity to read your interesting article.

I have just two suggestions for your consideration:

- Table 2 should be better explained;

With regard to table 3, although it is well constructed, I believe that they would be better understood if there were a complementary description of the most significant results.

Best regards

Author Response

We would like to express our gratitude to reviewer 2 for the time in reviewing our manuscript and for providing us comments helpful to improve this manuscript quality. We have answered their concerns (all corrections were marked in red and/or change control).

--------------------

Reviewer’ note: Table 2 should be better explained. With regard to table 3, although it is well constructed, I believe that they would be better understood if there were a complementary description of the most significant results.

Authors’ response: Thank you very much for your suggestions. A more serious statistical analysis (Linear Mixed Model-random factor = group-class; and Bonferroni Post Hoc) has been performed. Likewise, the results have been modified and explained in more detail (see Statistical Analysis and Results sections).

Author Response

We would like to express our gratitude to reviewer 3 for the time in reviewing our manuscript and for providing us comments helpful to improve this manuscript quality. We have answered their concerns (all corrections were marked in red and/or change control).

--------------------

Reviewer’ note: Mage.

Authors’ response: Thank you very much for your suggestion. The term "Age" has been replaced by "Mage" (Lines 18 and 119).

--------------------

Reviewer’ note: It would be good to use – enter shortcut – abbreviation for the physical education in the text from now on e.g. (PE).

Authors’ response: We agree with you. The abbreviation PE has been used throughout the text.

--------------------

Reviewer’ note: It would be good to use shortcut – abbreviation for the physical activity from now on e.g. (PA).

Authors’ response: We agree with you. The abbreviation PA has been used throughout the text.

--------------------

Reviewer’ note: Please add here a reference for the SDT theory. Please use a shortcut – abbreviation for Self-Determination theory e.g. (SDT).

Authors’ response: We agree with you. The abbreviation SDT has been used throughout the text. In addition, the following reference has been inserted (Line 36).

Deci, E.L.; Ryan, R.M. Intrinsic motivation and self-determination in human behavior; Plenum Press: New York and London, 1985.

--------------------

Reviewer’ note: Please re-write these phrase e.g. from 1 (strongly disagree) to 5 (strongly agree).

Authors’ response: Based on your suggestion, the phrase has been re-written (Line 153).

--------------------

Reviewer’ note: To my opinion, the whole Statistical analysis could be in one paragraph.

Authors’ response: Thank you very much for your comment. The statistical analysis has been grouped in one paragraph (Lines 159 to 171).

--------------------

Reviewer’ note: Please report here the test used to explore data normal distribution.

Authors’ response: Thank you very much for your comment. The test used to explore the normal distribution of the data has been reported (Line 161).

--------------------

Reviewer’ note: If possible, please add also the TLI values for the CFA in Table 2.

Authors’ response: Based on your suggestion, TLI, AVE and CR values have been added in Table 2.

--------------------

Reviewer’ note: et al.

Authors’ response: Based on your suggestion, “surnames” have been replaced by “et al”.

--------------------

Reviewer’ note: Somewhere in this paragraph should discuss why autonomy had low correlation with sport adherence in this study, what other studies have found regarding autonomy and sport participation. Also, you have to explain here why autonomy was not a significant predictor to PA.

Authors’ response: Thank you very much for your comments. It has been discussed what autonomy had a low correlation with sport adherence in this study (Line 273 to 277), and what other studies have found in relation to autonomy and sport participation.

García-Romero, C.; Méndez-Giménez, A.; Cecchini-Estrada, J.A. Predictive role of 3x2 achievement goals on the need for autonomy in Physical Education. Sportis-Scientific Technical Journal of School Sport Physical Education and Psychomotricity 2020, 6, 2-17, doi:10.17979/sportis.2020.6.1.5799.

Valero-Valenzuela, A.; Manzano-Sánchez, D.; Moreno-Murcia, J.A.; León, D.A.H. Interpersonal Style of Coaching, Motivational Profiles and the Intention to be Physically Active in Young Athletes. Studia Psychologica 2019, 61, 110-119, doi:10.21909/sp.2019.02.776.

Reviewer 4 Report

Abstract

The abstract concludes by stating that "increased sport adherence will have health benefits." While this may be true in general, it is an overgeneralization to make this claim based solely on the findings of this study.

Finally, in the abstract of the paper, there is no mention of the either theoretical or practical implications of the study. This information should be provided in the abstract of the paper

Introduction

Although the study aims to analyze the influence of gender, sport experience, and teaching method on psychological variables, the research question is not explicitly stated. The introduction does not clearly state the research question that the study aims to answer, which could make it difficult for readers to understand the purpose of the research.

The literature review in this introduction focuses on the Self-Determination Theory, the Basic Psychological Needs Theory, and the Tactical Games Approach. While these are all relevant theories for understanding motivation and physical activity, the review is limited in scope and does not incorporate a broader range of literature on gender, sport experience, and teaching methods.

The introduction does not clearly state why this research is important or necessary. While it briefly mentions that the lack of interest in physical education classes is increasing and that the teaching method could be a factor, it does not provide a compelling argument for why this particular study is needed.

The introduction cites several previous studies that have investigated similar topics, but it does not critically evaluate the limitations or gaps in this research. It is important to acknowledge the limitations of previous research and explain how the current study will address these limitations.

Discussion

The discussion section mainly focuses on describing the results of the study rather than interpreting them in the context of previous research and theories. The authors could have discussed how their findings relate to the existing literature on self-determination theory, physical education, and sport adherence, and what implications they have for future research and practice.

Secondly, the authors use the phrase "partially accepted" several times to describe the results of their hypotheses, but they do not explain what this means in practice or what the implications are for the study. It would be helpful if they provided more detail about what the partial acceptance of a hypothesis entails and what factors might have contributed to the partial acceptance.

Finally, the limitations of the study are mentioned briefly and not discussed in detail. The authors could have expanded on how the limitations might have affected the results and what implications this has for the generalizability of the study.

Reviewer 5 Report

The topic raises interesting questions, the research concept is good and the hypotheses are well formulated.

The literature needs to be expanded.

A better fit with the journal needs to be developed. What constitutes a contribution to sustainability in the present research? Do the results contribute to sustainability in the social, educational, or health systems?

A more detailed explanation of the results is needed. The presentation of results is poor, the methodology needs improvement. More serious statistical analysis using the existing database is recommended.

In its present form, a major revision of the manuscript is recommended.
I do not recommend publication of the paper in its present form.

I suggest a moderate editing of the English

Author Response

We would like to express our gratitude to reviewer 5 for the time in reviewing our manuscript and for providing us comments helpful to improve this manuscript quality. We have answered their concerns (all corrections were marked in red and/or change control).

--------------------

Reviewer’ note: The literature needs to be expanded.

Authors’ response: The literature has been expanded:

García-Romero, C.; Méndez-Giménez, A.; Cecchini-Estrada, J.A. Predictive role of 3x2 achievement goals on the need for autonomy in Physical Education. Sportis-Scientific Technical Journal of School Sport Physical Education and Psychomotricity 2020, 6, 2-17, doi:10.17979/sportis.2020.6.1.5799.

Hair, J.F.; Black, W.C.; Babin, B.J.; Anderson, R.E.; Tatham, R.L. Multivariate Data Analysis, 7th Ed.; Pearson Education Limited: Upper Saddle River, New Jersey, 2009.

Metzler, M.W. Instructional Models for Physical Education, 3rd Ed.; Holcomb Hathaway: Scottsdale, AZ, USA, 2011.

Mitchell, S.A.; Oslin, J.L.; Griffin, L.L. Teaching Sport Concepts and Skills: A Tactical Games Approach for Ages 7 to 18, 3rd Ed.; Human Kinetics: Champaign, IL, USA, 2013.

Valero-Valenzuela, A.; Manzano-Sánchez, D.; Moreno-Murcia, J.A.; León, D.A.H. Interpersonal Style of Coaching, Motivational Profiles and the Intention to be Physically Active in Young Athletes. Studia Psychologica 2019, 61, 110-119, doi:10.21909/sp.2019.02.776.

--------------------

Reviewer’ note: A better fit with the journal needs to be developed. What constitutes a contribution to sustainability in the present research? Do the results contribute to sustainability in the social, educational, or health systems?

Authors’ response: Based on your questions, we have explained why this study contributes to the sustainability of the educational, social and health systems has been indicated (Lines 92 to 95; 292-294; 303-305).

--------------------

Reviewer’ note: A more detailed explanation of the results is needed. The presentation of results is poor, the methodology needs improvement. More serious statistical analysis using the existing database is recommended.

Authors’ response: Thank you very much for your suggestions. A more serious statistical analysis (Linear Mixed Model-random factor = group-class; and Bonferroni Post Hoc) has been performed. In addition, the procedures and results have been explained in more detail (see Statistical Analysis and Results sections).

Round 2

Reviewer 1 Report

Thank You for Your job, all changes You've made in the paper made it easier to read and understand (for me). Now I don't see any week points of the manuscript.

Reviewer 4 Report

I'm satisfied with the revision. Well done.